# Spatial and temporal variations of $CO_2$ mole fractions observed at Beijing, Xianghe and Xinglong in North China

Yang Yang[1,2,7], Minqiang Zhou[2,3,7], Ting Wang[2,7], Bo Yao[6], Pengfei Han[5], Denghui Ji[2,7], Wei Zhou[4,7], Yele Sun[4,7], Gengchen Wang[2,7], and Pucai Wang[2,7]

[1]Shanghai Ecological Forecasting and Remote Sensing Center, Shanghai, China
[2]Key Laboratory of Middle Atmosphere and Global Environment Observation (LAGEO), Institute of Atmospheric Physics, Chinese Academy of Sciences, Beijing, China
[3]Royal Belgian Institute for Space Aeronomy, Brussels, Belgium
[4]State Key Laboratory of Atmospheric Boundary Layer Physics and Atmospheric Chemistry (LAPC), Institute of Atmospheric Physics, Chinese Academy of Sciences, Beijing, China
[5]State Key Laboratory of Numerical Modeling for Atmospheric Sciences and Geophysical Fluid Dynamics (LASG), Institute of Atmospheric Physics, Chinese Academy of Sciences, Beijing, China
[6]The China Meteorological Administration, Meteorological Observation Centre
[7]University of Chinese Academy of Sciences, Beijing, China

**Correspondence:** Minqiang Zhou (minqiang.zhou@aeronomie.be); Ting Wang (wangting@mail.iap.ac.cn)

**Abstract.** Atmospheric $CO_2$ mole fractions are observed at Beijing (BJ), Xianghe (XH), and Xinglong (XL) in North China using Picarro G2301 Cavity Ring-Down Spectroscopy instruments. The measurement system is described comprehensively for the first time. The geo-distances among these three sites are within 200 km, but they have very different surrounding environments: BJ is inside the megacity; XH is in the suburban area; XL is in the countryside on a mountain. The mean and standard deviation of $CO_2$ mole fractions at BJ, XH, and XL between October 2018 and September 2019 are 448.4±12.8 ppm, 436.0±9.2 ppm and 420.6±8.2 ppm, respectively. The seasonal variations of $CO_2$ at these three sites are similar, with a maximum in winter and a minimum in summer, which is dominated by the terrestrial ecosystem. However, the seasonal variations of $CO_2$ at BJ and XH are more affected by human activities as compared to XL. By using $CO_2$ at XL as the background, $CO_2$ enhancements are observed simultaneously at BJ and XH. The diurnal variations of $CO_2$ are driven by the boundary layer height, photosynthesis and human activities at BJ, XH and XL. We also compare the $CO_2$ measurements at BJ, XH and XL with 5 urban sites in US, and it is found that the $CO_2$ mean concentration at BJ is the largest. Moreover, we address the impact of the wind on the $CO_2$ mole fractions at BJ and XL. This study provides an insight into the spatial and temporal variations of $CO_2$ mole fractions in North China.

## 1 Introduction

Carbon dioxide ($CO_2$) is the largest contributor to the total positive radiative forcing of the earth among anthropogenic gases. $CO_2$ has reached up to 140% relative to the pre-industrial level mainly due to fossil fuel combustion and land-use change (IPCC, 2013). The increase in $CO_2$ has led to an imbalance of 0.58 ±0.15 $Wm^{-2}$ in energy budget between 2005 and 2010 at the top of atmosphere (Hansen et al., 2011), resulting into changes in the atmospheric temperature, the sea level, and the hydrology.

Urban areas only take up around 2% of global land cover, while they emit more than 70% of $CO_2$ emissions from burning

fossil fuels (Churkina, 2016). According to Gao et al. (2018), $CO_2$ emissions in metropolitan regions increased continuously from 1985 to 2006. Dhakal (2009) showed that China's urbanization rate has already reached 40% in 2005 and it is predicted to reach up to the level of 60% in 2030. This kind of increase certainly demands large quantities of energy consuming, leading to a large amount of $CO_2$ emissions.

It is important to understand atmospheric $CO_2$ variations in urban, suburban and rural areas. Previous studies carried out in

urban areas, such as Phoenix, USA (Idso et al., 2013) and Copenhagen, Denmark (Soegaard and Møller-Jensen, 2003) show that $CO_2$ mole fractions are larger in the city center as compared to the outskirts, which is called "urban $CO_2$ dome". Various underlying surfaces, such as buildings, roads, trees, croplands, and grasslands cause complicate $CO_2$ characteristics (Cheng et al., 2018). George et al. (2007) pointed out that the horizontal gradients of $CO_2$ mole fractions among urban, suburban and rural areas are caused by different population densities and traffic volumes.

The Beijing-Tianjin-Hebei (BTH) area is an economically dynamic region, located in North China, with highly urbanized cities, suburban cities and rural areas (Figure 1). During the last two decades, the population in Beijing has increased from 13.64 million in 2000 to 21.54 million in 2018, the car amount increases from 1.04 million in 2000 to 5.74 million in 2018 (http:/data.stats.gov.cn/). In the BTH area, the major $CO_2$ emissions are coming from industry, residential emissions, power plant and transportation (Song et al., 2013; Feng et al., 2019). In order to reduce the carbon emissions, Beijing has adopted a

number of vehicle emission control strategies since the mid-1990s, for example, emission control on new and in-use vehicles, fuel quality improvements, alternative-fuel and advanced vehicles and public transport (Wu et al., 2011). During the China's 12th (2011-2015) and 13th (2016-2020) Five-Year Plan periods, comprehensive work programs have been implemented for energy conservation and emission reduction in Beijing. More recently, Beijing has also launched the short-term 'the three-year blue-sky defense battle of Beijing' between 2018 and 2020. Regional networks incorporated with high-accuracy $CO_2$

measurements can be used to retrieve carbon emissions and sinks in the horizontal gradient. The vertical gradient of $CO_2$ mole fractions can also be observed at several different heights at the same location (Bakwin et al., 1998).

To better understand the characteristics of $CO_2$ variations in the BTH area, 3 Cavity Ring-down Spectroscopy (CRDS) analyzers (Picarro G2301) within 200 km were installed at Beijing (BJ), Xianghe (XH), and Xinglong (XL). The three sites have very different surrounding environments: BJ is inside the megacity, XH is in the suburban area, and XL is in the countryside

on a mountain. The measurements between June 2018 and April 2020 at the three sites allow us to better understand the differences among the urban, suburban and rural sites about the seasonal, synoptic and diurnal variations of $CO_2$ mole fractions. Section 2 describes the site locations as well as the measurement system. The results and discussions are presented in Section 3. Finally, the conclusions are drawn in Section 4.

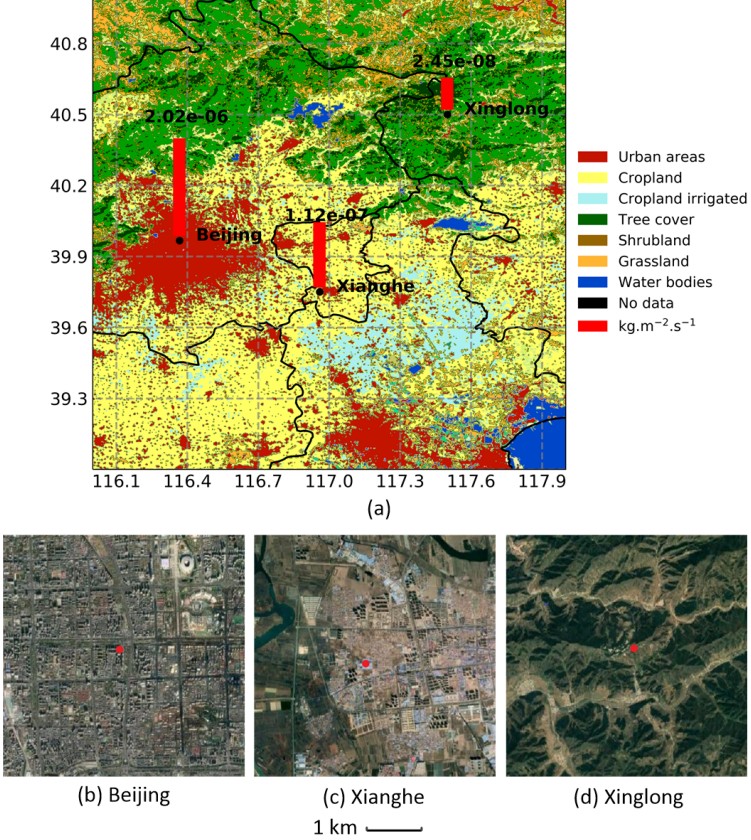

(a)

(b) Beijing  (c) Xianghe  (d) Xinglong

1 km ⸺

**Figure 1.** (a): the Location of three sites at Beijing (BJ, 39.96 °N, 116.36 °E, 49 m a.s.l.), Xianghe (XH, 39.75 °N, 116.96 °E, 30 m a.s.l.) and Xinglong (XL, 40.40°N, 117.50 °E, 940 m a.s.l.), together with the land cover in this area. The red bars are the carbon dioxide emissions at the 3 sites based on the EDGAR data. The map within ~2 km of BJ (b), XH (c) and XL (c) are coming from © Google Maps (https://www.google.com/maps).

## 2 Measurements

### 2.1 Sites

The locations of the three sites at BJ (39.96 °N, 116.36 °E, 49 m above sea level (a.s.l.)), XH (39.75 °N, 116.96 °E, 30 m a.s.l.) and XL (40.40°N, 117.50 °E, 940 m a.s.l.) are shown in Figure 1. The red bars above the sites are the anthropogenic carbon dioxide emissions in 2015 from the Emission Database for Global Atmospheric Research (EDGAR) v5.0 (Crippa et al., 2019). The $CO_2$ fluxes are $2.02 \times 10^{-6}$, $1.12 \times 10^{-7}$, and $2.45 \times 10^{-8}$ kg m$^{-2}$ s$^{-1}$ at BJ, XH and XL, respectively.

The BJ site is located in a highly urbanized area, with dense buildings, shopping centers, roads and residential districts. To the east of the site, there is the Beijing-Tibet expressway (G6) carrying a heavy volume of traffic. Within 1 km of the site,

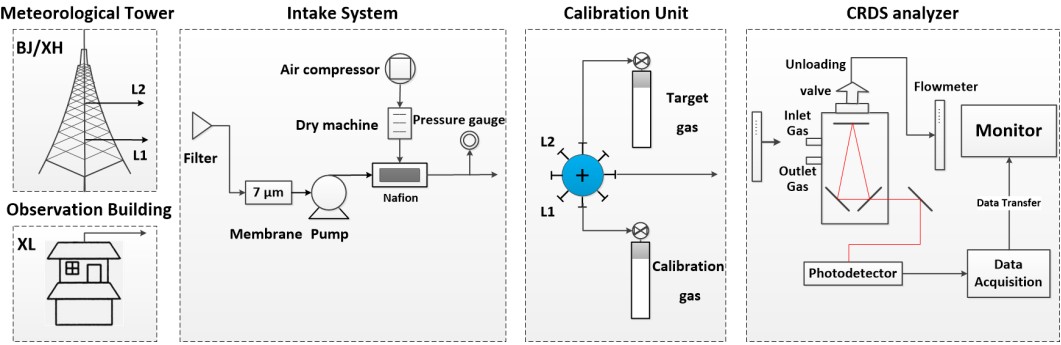

**Figure 2.** The schematic diagram of measurement system, including a meteorological tower at BJ/XH or observation building at XL, an intake system, a calibration unit and a CRDS analyzer.

the heights of trees are about 15-20 m, and the heights of buildings are about 70-200 m (Cheng et al., 2018). The vegetation fractions around the BJ site are between 10% and 18% (Liu et al., 2012).

The XH site is in a suburban area about 50 km to the southeast of Beijing. XH is surrounded by croplands and irrigated croplands. Within 1 km of the XH site, the residential houses are mainly home-built, with an average height of ~20 m. The center of Xianghe county is about 2 km to the east of the site.

The XL site is located on a mountain, inside the Xinglong Observatory of the National Astronomical Observatories, Chinese Academy of Sciences (NAOC) (https://www.xinglong-naoc.org/html/en/), which is about 120 km to the northeast of Beijing. XL is located in a highly vegetated area.

## 2.2   Measurement system

The Picarro Cavity Ring-Down Spectroscopy (CRDS) G2301 analyzers were installed at BJ, XH, and XL to measure $CO_2$, $CH_4$, and $H_2O$ mole fractions. The same measurement system is operated at these three sites, which is composed of an intake system, a calibration unit, and a Picarro analyzer (Figure 2). Note that there are two sampling heights at BJ (80 and 280 m above ground level (a.g.l.)) and XH (60 and 80 m a.g.l.), but only one sampling height (10 m a.g.l.) at XL. The measurements start in June 2018 at BJ and XH, and in May 2016 at XL. To compare the $CO_2$ measurements among these sites, we focus on the data after June 2018 in this study.

The surrounding air is sampled by a vacuum pump (DA7002D) with a maximum flux of 20 L.min$^{-1}$ through an inlet (Figure 2). The sample air is then introduced into a 10 mm-diameter tube (SYNFLEX 1300), mounted with a capsule filter (Whatman, USA) to filter out the solid particle with a diameter larger than 2 $\mu m$ and liquid particles with a diameter larger than 0.03 mm. In addition, a 7 mm sintered filter (Membrane) is installed to filter out the solid particle with a diameter larger than 7 $\mu m$. Moreover, an air compressor and a dry machine together with a single Nafion tubing selectively permeable membrane dryer (MD-110-72P-4; Perma Pure, Halma, UK) in self-purge are installed to remove water vapor. The sample dew-point temperature can reach down to -25 °C, corresponding to a relative humidity of 1-20 %. The flux of the Nafion outflow is 200-400 ml min$^{-1}$.

The outflow is then vented to the unloading valve (Figure 2), which guarantees that the air fed to the Picarro G2301 analyzer is controlled at near-ambient pressure. Before the ambient $CO_2$ measurements, the sampled air is introduced to the calibration unit to check the precision and stability of the system, which will be introduced in detail in Section 2.3.

The last part of the measurement system is the Picarro analyzer, which is composed of a laser, a high-finesse optical cavity, and a detector. The sample air is first introduced into the cavity. After that, the laser passes through the sample air and the intensity of the laser arriving at the detector is monitored as $I$. Then, the 'ring-down' measurements start as the laser rapidly shuts down. Meanwhile, the sample gas is measured by recording the decay of the laser intensity with time. This decay depends on the optical path inside of the cavity, which is in correlation with the absorption and scattering of the sample air. The analyzer continuously scans the laser over $CO_2$ spectral features and records the absorption loss at a wavelength of 1603 nm to form the spectrum. As a result, $CO_2$ mole fractions are derived from these spectra and collected by the Data Acquisition part.

### 2.3 Calibration method

As is shown in Figure 2, the intake system is connected to an 8-position valve, which is used to choose the air coming from the sample air, the target gas, or the calibration gas. The target and calibration gases are pressurized in 29.5 L treated aluminum alloy cylinders, which are scaled to the WMO X2007 standard by the China Meteorological Administration, Meteorological Observation Centre. The same calibration procedure is operated at these three sites: 1) 3-hours sample air; 2) 5-minutes calibration gas; 3) 3-hours sample air; 4) 5-minutes target gas. This process repeats every 6 hours and 10 minutes. Note that, the airs coming from two levels at XH and BJ are switched every 5 minutes during the 3-hours sample air period. As the remaining volume in the tubes needs time for flushing, the response of the analyzer turns to be stable about 1 minute after each switching. In order to reduce the uncertainty, we do not consider the first 3-minutes measurements after each switching.

The calibration gas is to calculate the calibration factor ($cf$),

$$cf = \overline{CO_{2,mcal}/CO_{2,cal}}, \tag{1}$$

where $CO_{2,mcal}$ is the $CO_2$ mole fraction measured by the Picarro analyzer from the calibration gas and $CO_{2,cal}$ is the standard $CO_2$ mole fraction of the calibration cylinder. $cf$ is applied to correct the sample air during the next 6 hours,

$$CO_{2,c} = cf \times CO_{2,m}, \tag{2}$$

where $CO_{2,m}$ is the $CO_2$ mole fraction measured by the Picarro analyzer and $CO_{2,c}$ is the calibrated $CO_2$ mole fraction.

The target gas is used to check the precision and stability of the system. The T value are calculated as follows,

$$T = cf \times CO_{2,mtar} - CO_{2,tar}, \tag{3}$$

where $CO_{2,tar}$ is the standard $CO_2$ mole fraction of the target gas cylinder, $CO_{2,mtar}$ is the $CO_2$ mole fraction measured by the Picarro analyzer from the target gas.

To keep the CRDS stable over time, only the periods with T value within $\pm$ 0.1 ppm are selected (Fang et al., 2014). The measurement uncertainties of the Picarro instrument at the three sites are calculated as the standard deviation (std) of T, which are 0.01, 0.06, and 0.02 ppm at BJ, XH, and XL respectively.

## 2.4 Data quality control

Besides the calibration procedure mentioned in Section 2.3, we also do auto and manual flagging of the raw data. In each 1-hour $CO_2$ measurement window, auto-flags are assigned when deviations from $CO_2$ mean are found larger than 2-times hourly $CO_2$ std. Furthermore, manual flags are assigned by technicians at each site according to the logbook to exclude no-valid data resulted from the inlet filter, pump, and extreme weather issues. In addition, as the CRDS measurement system records $CO_2$ and $CH_4$ simultaneously, the variations of these two gases are checked together to manually flag $CO_2/CH_4$ outliers.

## 2.5 Meteorological fields

The $CO_2$ variations are additionally characterized by specific meteorological parameters, such as local wind and temperature fields. The meteorological sensors at BJ are installed at the same tower as the Picarro on 120 m a.g.l., and the meteorological sensors at XL are ~5 m northwest to the Picarro sample tube. The meteorological fields at XH are not discussed here as there is a technical issue with the wind sensor.

Figure 3 shows the wind frequencies at BJ and XL in each season, which are binned with a resolution of 2 m.s$^{-1}$ for the wind speed and 10° for the wind direction. At BJ, two dominant wind regimes are observed throughout the whole year: north (northwest to northeast clockwise) and southwest. The percentage of wind frequency in the north region is 34%, 36%, 50% and 60% respectively from spring to winter. The wind speed varies from 0.63 m.s$^{-1}$ on 10 May 2019 to 14.98 m.s$^{-1}$ on 20 December 2018, with a mean of 3.92 m.s$^{-1}$. From spring to autumn, more winds are with a low wind speed. However, in winter, the prevailing northwest wind contributes to high wind frequencies with the increase of wind speed. At XL, the dominant winds are mainly from the west (southwest to northwest clockwise), together with some winds from the southeast. The percentage of wind frequency in the west region is 52%, 33%, 56% and 57% respectively from spring to winter. The wind speed varies from near-zero on 18 August 2019 to 10.75 m.s$^{-1}$ on 17 April 2019, with a mean of 2.52 m.s$^{-1}$.

The atmospheric boundary layer height (BLH) is another important parameter to characterize the diurnal variation of $CO_2$ (Li et al., 2014; Culf et al., 1997). In this study, we use the BLH hourly data of the ERA5 reanalysis data from the European Centre for Medium-Range Weather Forecasts (ECMWF) with a spatial resolution of 0.25 °× 0.25 °(Hersbach et al., 2020).

## 3 Results and discussions

### 3.1 $CO_2$ time series and comparison with other urban sites

Figure 4 shows the time series of hourly $CO_2$ mole fractions at the three sites between June 2018 and March 2020. The two-levels (80 m and 280 m) measurements at BJ are marked as BJ L1 and BJ L2, and the two-levels (60 m and 80 m) measurements at XH are marked as XH L1 and XH L2. The gaps in the $CO_2$ time series are due to the malfunctions of the instruments. To better understand the influence of the wind on $CO_2$, we classify the $CO_2$ mole fractions at XL and BJ L1 based on the wind information into five classes respectively (Fig. 4a and b). The BJ L1 is used here as it is closer to the wind sensor as compared

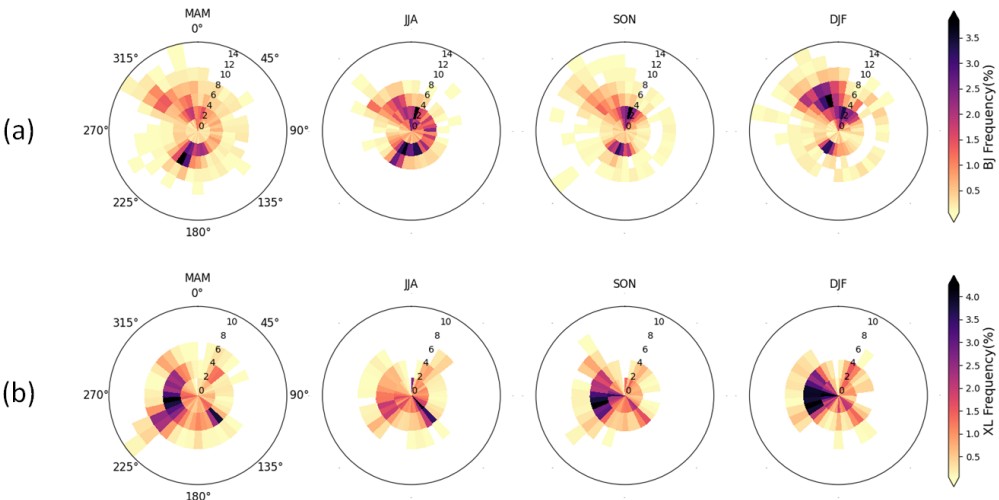

**Figure 3.** Wind frequency as a function of wind speed (m.s$^{-1}$) and wind direction (°) in spring (MAM), summer (JJA), autumn (SON) and winter (DJF) at BJ (a) and XL (b) from October 2018 to September 2019.

to BJ L2. The local class is defined as wind speed less than 2 m.s$^{-1}$, while the wind speed larger than 2 m.s$^{-1}$ are classified into four sections according to the wind direction: northwest (NW), northeast (NE), southwest (SW) and southeast (SE).

As expected, the urban BJ site observes a much higher $CO_2$ level than the suburban XH and rural XL sites. The $CO_2$ measurements at the urban site BJ L1 (Fig. 4b) are influenced by the wind speeds and wind directions. High $CO_2$ mole
fractions generally appear in local class throughout the whole year, indicating the strong local anthropogenic emissions. The north sectors (NS and NE) usually contribute low $CO_2$ mole fractions during the autumn-winter period. However, in spring and summer, the SW sector contributes lower $CO_2$, indicating the low $CO_2$ varies with the wind direction season by season at BJ. Different from BJ, the $CO_2$ mole fraction in the local class at XL covers all the data range throughout the whole year. In spring and summer, the wind from the south (SE and SW) makes $CO_2$ increase at XL.

Comparisons with other five urban sites in USA with a similar latitude of BJ are also discussed in this section. All these five sites belong to the $CO_2$ Urban Synthesis and Analysis (CO2-USA) Data Synthesis Network (Feng et al., 2016). The site locations, elevations, inlet heights, and references are listed in Table 1. As the $CO_2$ measurements at these five sites do not cover the period between October 2018 and September 2019, we use the latest 1-year available $CO_2$ measurements. The monthly means and diurnal cycles of $CO_2$ at BJ (L1), XH (L1), and 5 American urban sites are shown in Figure 5. It is found that
the phases of the seasonal $CO_2$ cycles at BU, CRA, COM, IMC and SF are consistent with the observations at BJ (L1), XH (L1) and XL, with a high value in autumn-winter and a low value in summer. Among the five American sites, the highest $CO_2$ concentration is observed at IMC. The IMC site is inside a commercial zone and the $CO_2$ measurements over there are more strongly influenced by local emissions over there (Bares et al., 2019). The $CO_2$ concentration is also high at COM, because the Los Angeles megacity is one of the largest fossil fuel $CO_2$ emitters in the world (Matthäus et al., 2021). Figure 5 (a) shows that

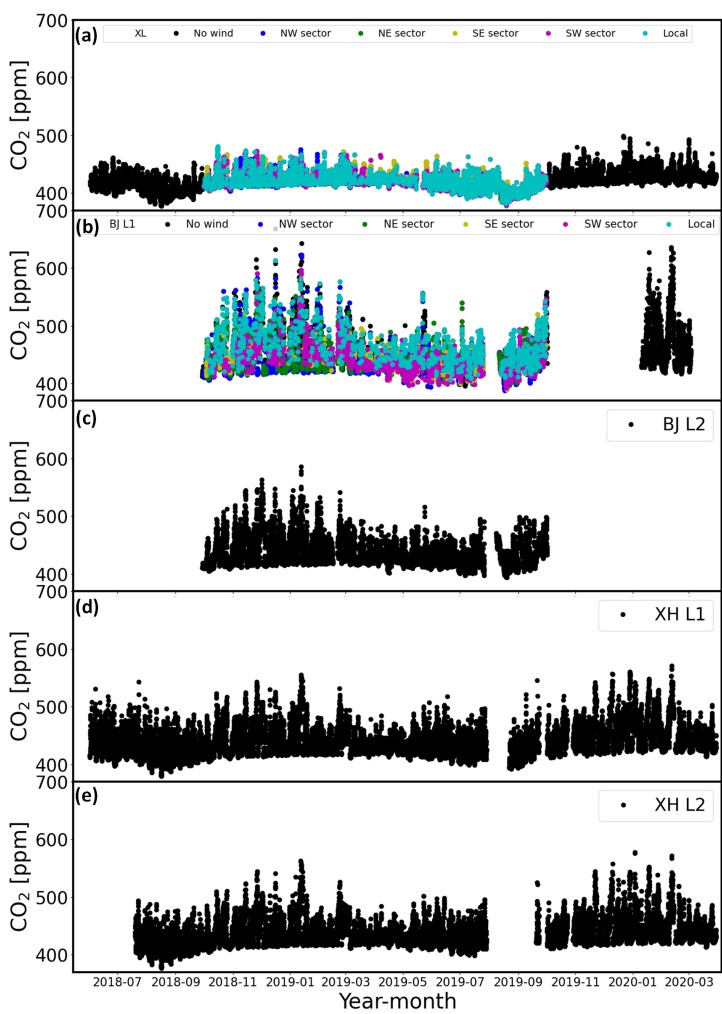

**Figure 4.** The time series of the CO$_2$ measurements at XL (a), BJ L1 (b), BJ L2 (c), XH L1 (d) and XH L2 (e) between June 2018 and March 2020. The CO$_2$ measurements at XL (a) and BJ LI (b) are colored by wind classes discussed in the text.

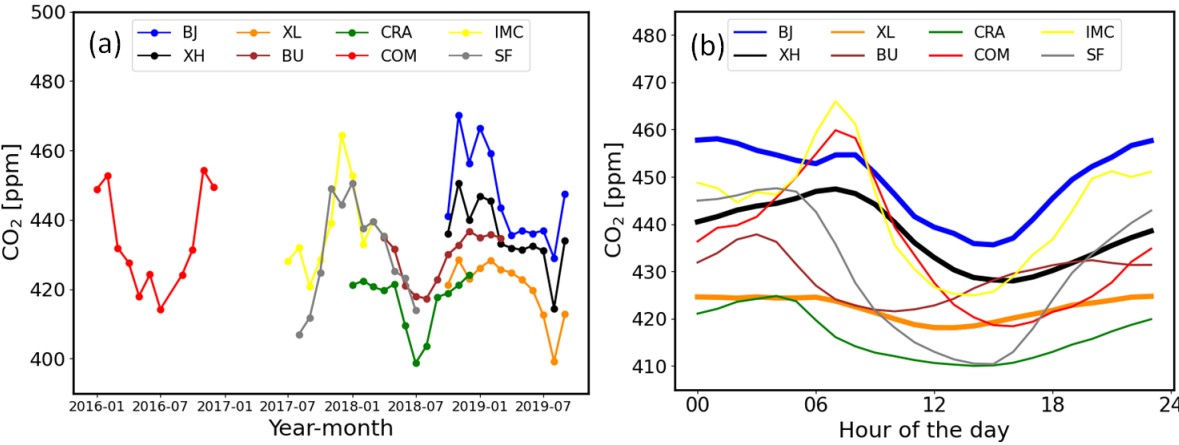

**Figure 5.** (a) Monthly means of $CO_2$ at BJ (L1), XH (L1), XL between October 2018 and September 2019, at BU, CRA, COM, IMC and SF during the latest 1 year and (b) the diurnal cycles of $CO_2$.

the $CO_2$ concentrations at COM and IMC are in the same level with the one at XH, but are less than the $CO_2$ concentration at BJ. The $CO_2$ concentrations at SF, BU and CRA are much lower as compared to BJ, because of lower anthropogenic emissions at these sites (McKain et al., 2015; Lauvaux et al., 2016; Shusterman et al., 2016).

Figure 5 (b) shows the diurnal variations of $CO_2$, with the amplitudes of 22.4, 19.4, 6.6, 16.3, 14.8, 41.5, 41.1 and 37.2 ppm at BJ (L1), XH (L1), XL, BU, CRA, COM, IMC and SF, respectively. The amplitudes of the diurnal variation at COM, IMC and SF are higher than that at BJ, although the yearly mean $CO_2$ levels at these sites are smaller than that at BJ. As the sampling heights at these sites and BJ are similar, the large amplitudes of the diurnal variation indicate that stronger variation in the local emissions and/or sinks exists at these three American sites as compared to BJ.

### 3.2 Contribution of main $CO_2$ sources

We use the CarbonTracker model, version CT-NRT.v2021-3 (Peters et al., 2005) to evaluate the influence of anthropogenic, biogenic, oceanic and fire sources at these three sites respectively. The CarbonTracker is a data assimilation system developed by the National Oceanic and Atmospheric Administration (NOAA) to keep track of sources and sinks of atmospheric $CO_2$ around the world. Four tracers (biosphere, ocean, fire and fossil fuel) are treated separately to simulate atmospheric $CO_2$ mole fractions. Mustafa et al. (2020) evaluated the CarbonTracker model in Asia by comparing with satellite measurements, and they found that the CarbonTracker model captures the variation of $CO_2$ well. The model provides 3-hourly $CO_2$ data at 25 levels from surface to ~123 km, and the spatial resolution of the global CarbonTracker $CO_2$ simulation is $3° \times 2°$(longitude $\times$ latitude). As BJ and XH are in the same model grid, we note the $CO_2$ simulations in the BJ/XH grid as BJ.

Figure 6 shows the time series of $CO_2$ simulations from fossil fuel ($CO_{2,ff}$), biosphere ($CO_{2,bio}$), fire ($CO_{2,fire}$) and ocean ($CO_{2,oce}$) modules at BJ/XH and XL between October 2018 and September 2019. It is found that the fire and ocean $CO_2$ at BJ/XH and XL are close to each other throughout the whole year. According to the Global Fire Assimilation System (GFAS)

**Table 1.** Site characteristics of BJ, XH and XL in North China, BU, CRA, COM, IMC and SF in USA from the CO2 Urban Synthesis and Analysis (CO2-USA) Data Synthesis Network.

| Site Code | Site Name | Lat (°N) | Lon (°E) | Elevation (m a.s.l.) | Inlet Height (m a.g.l.) | City | Reference |
|---|---|---|---|---|---|---|---|
| BJ | Beijing | 39.96 | 116.36 | 49 | 80/280 | Beijing | Cheng et al. (2018) |
| XH | Xianghe | 39.75 | 116.96 | 30 | 60/80 | Xianghe | Yang et al. (2020) |
| XL | Xinglong | 40.40 | 117.50 | 940 | 10 | Xinglong | Yang et al. (2019) |
| BU | Boston University | 42.35 | -71.10 | 4 | 29 | Boston | Sargent et al. (2018) McKain et al. (2015) |
| CRA | Crawfordsville | 39.99 | -86.74 | 264 | 76 | Indianapolis | Lauvaux et al. (2016) Richardson et al. (2017) |
| COM | Compton | 33.87 | -118.28 | 9 | 45 | Los Angeles | Verhulst et al. (2017) |
| IMC | Intermountain Medical Center | 40.67 | -111.89 | 1316 | 66 | Salt Lake City | Mitchell et al. (2018) Bares et al. (2019) |
| SF | SF Hospital Bldg 5 | 37.76 | -122.41 | 23.9 | 52 | San Francisco | Shusterman et al. (2016) |

180 (https://www.ecmwf.int/en/forecasts/dataset/global-fire-assimilation-system/) wildfire emissions, there is almost no biomass burning $CO_2$ emissions at BJ, XH and XL sites. The CarbonTraker model simulations confirm that fire $CO_2$ concentrations in this region are almost the same, and the simulated fire $CO_2$ at these sites are transported by the wildfire emissions at other places. What's more, the CarbonTraker model suggests that the fire $CO_2$ at these sites only take up a small proportion of the observed $CO_2$ (less than 5%). The biogenic $CO_2$ at BJ/XH and XL have a similar level between October 2018 and

185 June 2019, and become slightly different in summer 2019. However the difference in biogenic $CO_2$ is much less than that of the anthropogenic $CO_2$ differences. The high $CO_2$ concentrations at BJ and XH in winter are apparently dominated by the enhancement of fossil fuel. The variation of the fossil fuel $CO_2$ at XL is much less than that at BJ/XH. Therefore, by using the $CO_2$ measurements at XL as the background, we can significantly reduce the influence from fire, biosphere and ocean, and extract the signal of the anthropogenic $CO_2$ differences.

190  The $CO_2$ enhancement at BJ or XH relative to XL is then calculated as

$$\Delta CO_{2,BJ/XH} = CO_{2,BJ/XH} - CO_{2,XL} \tag{4}$$

The time series of hourly $\Delta CO_{2,BJ/XH}$ are presented in Figure 7a. The $\Delta CO_2$ has a maximum in winter and a minimum in summer at both BJ and XH. The high value is probably related to more combustion of fossil fuel from traffic and heating systems in winter (Liu et al., 2012). The daily $\Delta CO_2$ can reach up to 106.8 ppm in December 2018 at BJ and 78.5 ppm in

195 January 2019 at XH. The mean $\Delta CO_2$ at BJ and XH are $26.2 \pm 20.6$ ppm and $15.2 \pm 13.6$ ppm, respectively. There are 271 days when $\Delta CO_2$ are observed at both BJ and XH (Figure 7b). The correlation efficiency (R) of 0.81 is found between the $\Delta CO_2$ at BJ and XH, indicating that the $\Delta CO_2$s change simultaneously at BJ and XH. The slope of the linear fitting suggests that the $\Delta CO_2$ at BJ is 1.23 times larger than that of XH.

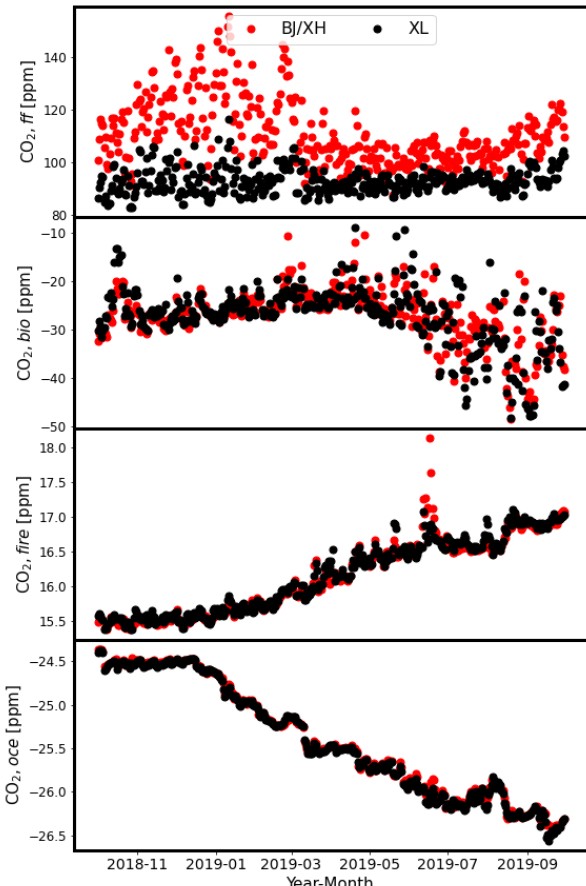

**Figure 6.** The time series of $CO_2$ simulations from fossil fuel ($CO_{2,ff}$), biosphere ($CO_{2,bio}$), fire ($CO_{2,fire}$) and ocean ($CO_{2,oce}$) modules at BJ/XH and XL.

## 3.3 Seasonal variations

The seasonal cycles of $CO_2$ are derived from the measurements at the lower levels at BJ and XH, and the measurements at XL. The lower levels at BJ and XH are used here as they reflect more information about surface fluxes. Figure 8a shows the $CO_2$ monthly means between October 2018 and September 2019, together with the temperature at BJ and leaf area index (LAI). The LAI monthly data are from the Copernicus Global Land Service (https://land.copernicus.eu/global/products/lai) with a spatial resolution of 1 km. Figure 8a shows the LAI monthly means in the region of Fig. 1.

Between October 2018 and September 2019, the mean of $CO_2$ mole fractions at BJ is 448.4±12.8 ppm, which is larger than those at XH (436.0±9.2 ppm) and XL (420.6±8.2 ppm). The phases of the seasonal cycle of $CO_2$ at BJ, XH and XL are similar, with a high value in autumn-winter and a low value in summer, which is consistent with other observations in North Hemisphere (Nevison et al., 2008). It is expected mainly due to the seasonal cycle of the biosphere fluxes (LAI). The increased

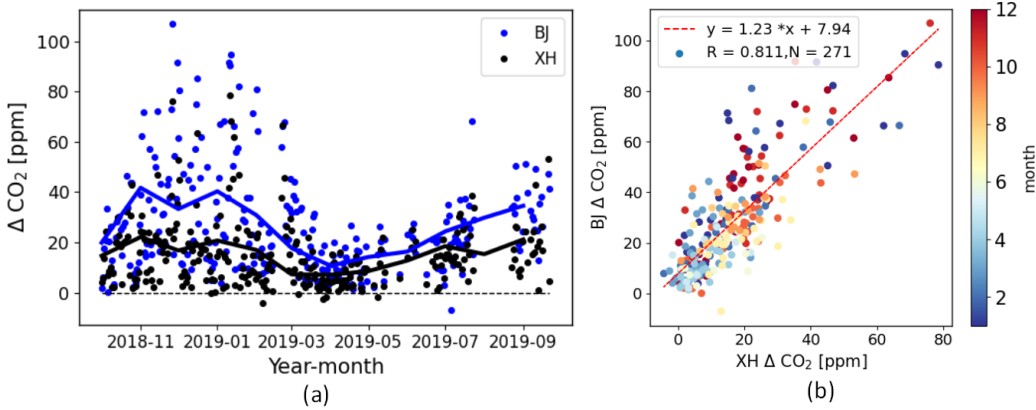

**Figure 7.** (a): the time series of daily $CO_2$ enhancements at BJ and XH relative to XL between October 2018 and September 2019. The blue and black lines are the monthly means of $CO_2$ enhancements at BJ and XH, respectively. (b): the correlation between daily mean $CO_2$ enhancements at BJ and XH.

temperature in summer is favorable for plant growth, leading to larger photosynthesis. In winter, the respiration of plants and
the anthropogenic heating emissions contribute to a high $CO_2$ level. The amplitudes of the seasonal variation of $CO_2$ at BJ, XH and XL are 41.2 ppm, 36.1 ppm and 29.3 ppm, respectively. According to the CarbonTracker simulation (Figure 6), the $CO_2$ seasonal cycle in this region is mainly driven by the biogenic and anthropogenic $CO_2$. At XL, the anthropogenic $CO_2$ is almost constant through the whole year, while the biogenic $CO_2$ is low in summer and high in winter. For BJ/XH, apart from the similar biogenic $CO_2$ seasonal variation, the anthropogenic $CO_2$ is also high in winter and lower in summer. Therefore,
combining the effect from the biosphere and human activities, the amplitude of $CO_2$ seasonal variation at BJ/XH is larger than that at XL. What's more, as the anthropogenic emission at BJ is much larger than that at XH, indicated by the EDGAR emission dataset, we thus observe the largest amplitude of the seasonal variation at BJ.

     Figure 8b, c and d show the $CO_2$ monthly means together with the monthly $1\sigma$ standard deviation at each site. We take the days when measurements are available at all three sites or the days when measurements are available at XH and XL. The $CO_2$
variability ($1\sigma$) is highest at BJ and lowest at XL. The seasonal $CO_2$ variation and $1\sigma$ standard deviation at each site are further assessed in the following.

     *Autumn*. At each site, monthly mean $CO_2$ mole fractions are increasing with the decrease of LAI. The increase rates of $CO_2$ at BJ, XH and XL are 30, 19 and 9 ppm/month, respectively. The $1\sigma$ standard deviation of each month at BJ is generally larger than that of XH, then followed by XL.

*Winter*. The $CO_2$ removed by the photosynthesis is weak in this region as the LAI is low. The $CO_2$ change simultaneously at BJ and XH, increasing from December 2018 to January 2019 and decreasing afterwards. Similar to autumn, the month-to-month variation of $CO_2$ at BJ is larger than those at BJ and XL, together with the largest $1\sigma$ at BJ. The $1\sigma$ at BJ and XH is larger in winter as compared to other seasons.

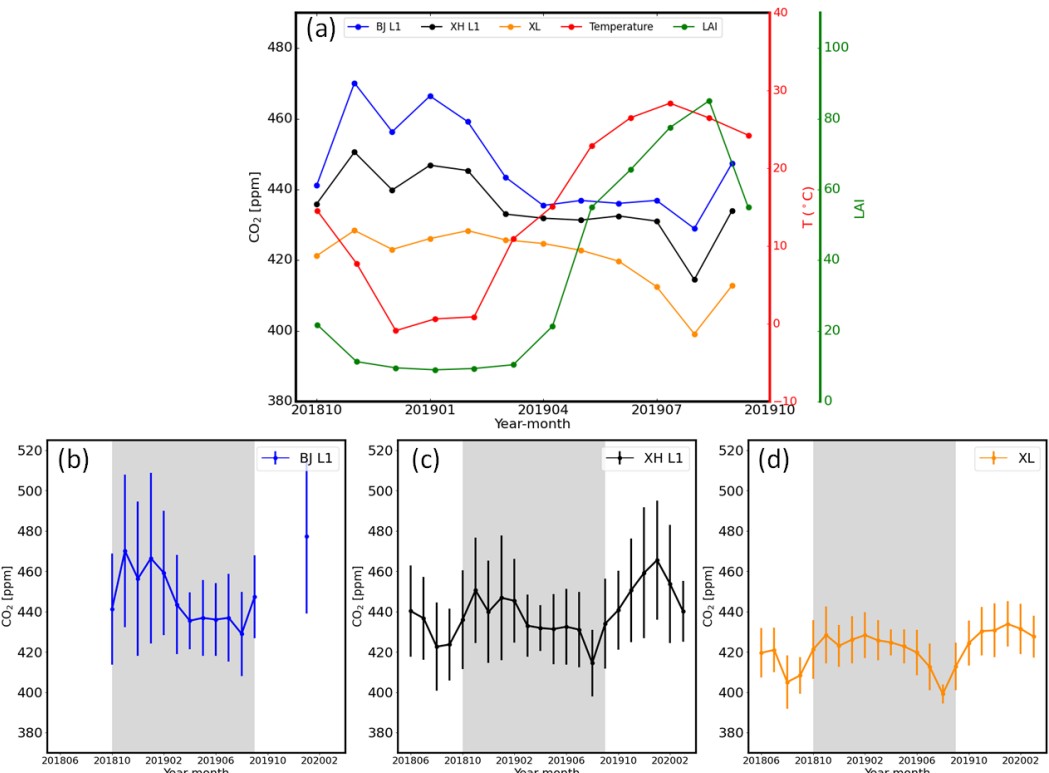

**Figure 8.** (a): the monthly means of $CO_2$ at BJ L1, XH L1 and XL between October 2018 and September 2019. The monthly mean air temperature at BJ and regional mean Leaf Area Index (LAI) of the area in Figure 1a during the same period are also displayed. (b-d): monthly means of $CO_2$ together with the $1\sigma$ standard deviation at BJ L1, XH L1 and XL between June 2018 and February 2020. The gap at BJ L1 is due to the instrument failure. The shadow is the measurement period displayed in Figure 8a.

*Spring*. The decrease of $CO_2$ in March 2019 is highly related to the temperature increase. As the heating is officially stopped in the middle of March, the anthropogenic emissions are much reduced (Shi et al., 2020). In April and May, the LAI increases significantly, leading to the decrease of $CO_2$, especially at XL. The regional biosphere activity affects more on $CO_2$ mole fractions at XL, while the large anthropogenic emissions at BJ and XH may reduce the influence from the photosynthesis.

*Summer*. At all the sites, the minimum $CO_2$ is observed in August with the maximum LAI corresponding to the largest photosynthesis $CO_2$ absorption activity. The month-to-month variation of $1\sigma$ is small at BJ and XH.

## 3.4 Diurnal variations

The diurnal variations of $CO_2$ at BJ, XH and XL between October 2018 and September 2019 are shown in Figure 9. The amplitudes of the diurnal variations are between 16.4 ppm and 44.1 ppm at BJ. The relatively large amplitudes are observed in summer and winter compared to spring and autumn. The phase of the diurnal variation at BJ varies with season. There are one

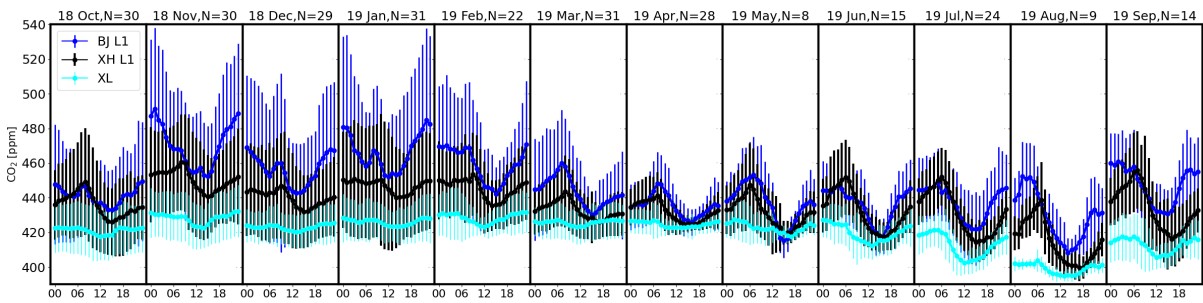

**Figure 9.** The diurnal cycles of $CO_2$ variations at BJ L1, XH L1 and XL in each month between October 2018 and September 2019. The collocated days are displayed (N). The error bars are the hourly standard deviations of $CO_2$.

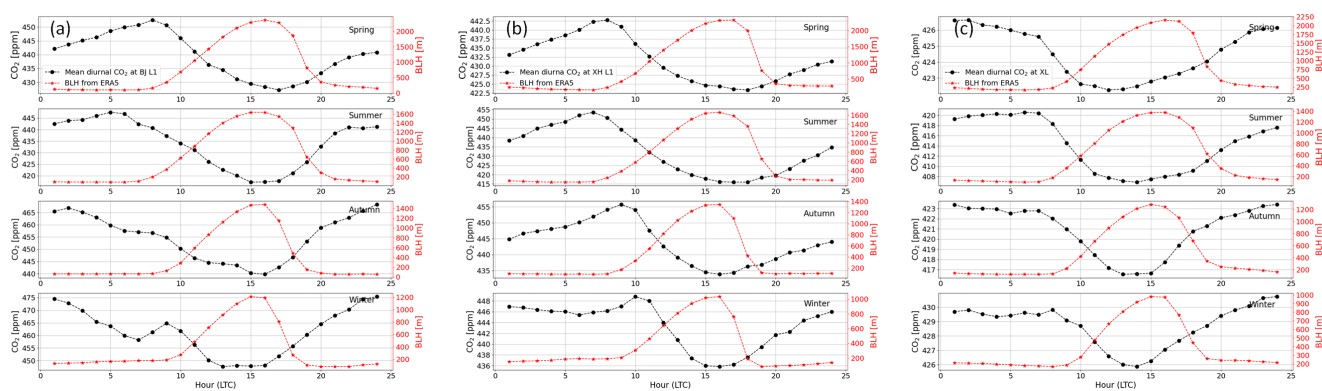

**Figure 10.** (a-c): mean diurnal cycles of BLH from ERA5 and mean diurnal $CO_2$ variations at BJ L1 (a), XH L1 (b) and XL (c) in each season between October 2018 and September 2019.

peak in the early morning (4:00-7:00) and one trough in the afternoon (14:00-16:00) in spring and summer. However, there are two peaks (8:00-9:00, 22:00-1:00), and two troughs (4:00-7:00, 14:00-16:00) in late autumn and winter. At XH, there are one peak (4:00-7:00) and one trough (14:00-16:00) throughout the whole year. The amplitude of the diurnal variation at XH is about 6-20 ppm smaller than that at BJ between November 2018 and May 2019. At XL, the peak of $CO_2$ occurs around 4:00-7:00, and the trough occurs in the afternoon around 12:00-14:00. The amplitudes of diurnal variations at XL are larger in summer as compared to other seasons. Moreover, the amplitudes of diurnal variations at XL are much smaller as compared to those at BJ and XH, especially in winter.

The diurnal variations of $CO_2$ are mainly affected by the BLH, photosynthesis, and local human activities (Chan et al., 2008; Denning et al., 1999). Generally, the increase of sunlight enhances the plant photosynthetic rate, vice versa. There is no photosynthetic $CO_2$ sink before sunrise or after sunset (Lv et al., 2020; Bagley et al., 2015). To better understand the influence

of the BLH on the diurnal $CO_2$ variations, we show the $CO_2$ diurnal cycles for each season at BJ L1, XH L1 and XL, together with the BLH hourly means.

**BJ L1**. The increase of the BLH after sunrise (5:00 - 8:00) and the photosynthetic uptake during the day make the $CO_2$ mole fraction decrease. The $CO_2$ mole fraction reaches a minimum in the afternoon around 16:00-17:00, corresponding to the maximum BLH. After that, the BLH decreases resulting into the accumulation of $CO_2$. In spring and summer, the $CO_2$ mole fraction keeps increasing until the next day (5:00-8:00) before sunrise, and in autumn and winter, the $CO_2$ mole fraction starts decreasing at midnight. Note that the enhancement of $CO_2$ around 9:00 in winter is not related to the BLH, which is probably due to the rush traffic emission.

**XH L1**. Similar to BJ, the variation of the $CO_2$ mole fraction is dominated by the BLH during the day. The $CO_2$ mole fraction decreases with the increase of BLH. The $CO_2$ mole fraction reaches a minimum in the afternoon around 16:00-17:00, corresponding to the highest BLH. However, at night, the variation of $CO_2$ at XH is not the same as that at BJ, especially in autumn and winter. In autumn, the $CO_2$ mole fraction keeps increasing until next day before sunrise (5:00-8:00), and in winter, the $CO_2$ mole fraction stays stable after midnight. Similar to BJ, the peak $CO_2$ around 9:00-10:00 in winter may be due to the traffic emission in the rush hour.

**XL**. Different from BJ and XH, the minimum of the $CO_2$ mole fraction occurs earlier than the maximum of BLH in spring and summer. For example, the minimum of the $CO_2$ mole fraction is around 12:00 and the maximum of BLH occurs around 16:00. The solar radiation is strongest at noon, which leads to the largest photosynthesis removing $CO_2$. The diurnal variation of $CO_2$ at daytime is then strongly affected by the plants in these two seasons. However, in autumn and winter, the minimum of the $CO_2$ mole fraction occurs close to the maximum of the BLH, which is also dominated by the change of PBL due to the low LAI in these two seasons (Mohotti and Lawlor, 2002; Newman et al., 2013).

### 3.5 $CO_2$ variations with the wind

Wind speed and wind direction are the two key factors in modulating the dispersion of $CO_2$ emissions (Turnbull et al., 2015; Lac et al., 2013; ángeles García et al., 2012). The influence of wind on $CO_2$ mole fraction at BJ and XL is discussed specifically in this section. To minimize the influence from the diurnal variation, we focus on the measurements between 14:00 and 16:00 during daytime for the highest BLH, and between 22:00 and 02:00 during nighttime for the lowest BLH. Besides, we reduce the impact from the seasonal variation of $CO_2$ by applying the following method. First, we calculate the mean of $CO_2$ over 10 days ($CO_{2,10d}$). Second, the ratio between the $CO_{2,10d}$ and the annual mean of original $CO_2$ is derived ($Index_{10d} = CO_{2,10d}/CO_{2,mean}$), and the $Index_h$ is interpolated from the $Index_{10d}$ at an hourly scale. Finally, The deseasonalized $CO_2$ is calculated as $CO_{2,de} = CO_2/Index_h$. In summary, we use the deseasonalized $CO_2$ during the daytime (14:00-16:00) and the nighttime (22:00-02:00) separately to understand the influence of the wind.

Figure 11 shows the daytime and nighttime wind roses of $CO_2$ mole fractions at BJ and XL, with a resolution of 1 m.s$^{-1}$ wind speed and of 10 °wind direction. Note that only the bins with the measurement number larger than 3 at BJ or 5 at XL are shown here.

*BJ*. At BJ, the wind mainly comes from the southwest and the northwest, with more winds come from the southwest during the day and more winds come from the northwest at night. The high $CO_2$ mole fractions are observed with a low wind speed ($<2$ m.s$^{-1}$). For the wind with a relatively large speed ($>2$ m.s$^{-1}$), it is found that the $CO_2$ with the wind coming from the southwest is about ~21 ppm larger than those with the wind coming from the northwest during the day.

*XL*. The wind speed at XL is generally smaller as compared to BJ. The wind at XL is mainly coming from the southeast-northwest sector in a clockwise direction. During the day, the high $CO_2$ mole fractions are observed along with a relatively large wind speed ($>2$ m.s$^{-1}$). This can be attributed to the impact of remote emissions advocated from the south, where large cities, such as Beijing and Tianjin, are located. At night, although the dominant wind shifts to the west, the high $CO_2$ mole fractions can be observed in almost all the directions with wind speeds ranging from 0 to 3 m.s$^{-1}$.

## 3.6 Two-levels measurements at BJ and XH

Figure 12 shows the $CO_2$ hourly means observed at two levels at BJ and XH between October 2018 and September 2019. Note that, we select measurements when the hourly means are available at both levels.

At BJ, $CO_2$ mole fractions at L1 are generally higher than L2 as L1 is closer to near-ground human emissions. At BJ L1 (80 m a.g.l.), we can observe a peak in the early morning, which is corresponding to the transportation rush hour. The valley of $CO_2$ at BJ L1 occurs at 16:00-17:00 because of the maximum PBL resulting from the unstable atmosphere. After that, the atmosphere changes from unstable to stable during the night, leading to the $CO_2$ peak again. At BJ L2 (280 m a.g.l.), the diurnal variation of $CO_2$ generally follows that at L1. Note that the peak of the $CO_2$ at L2 occurs in the early morning later than that at L1 as the $CO_2$ at the ground level moved upward with the increase in convective PBL, with a large difference in winter and a small difference in summer. The $CO_2$ diurnal variations from two-layers Picarro measurements in 2018 and 2019 in our study are consistent with the seven open-path infrared gas analyzers (Model LI-7500A; at 8, 16, 47, 80, 140, 200 and 280 m a.g.l.) measurements between 2013 and 2016 at the same site (Cheng et al., 2018). In summer, the temperature is high due to a larger solar irradiance, the atmosphere becomes unstable quickly accelerating the uplifting of the PBL. In winter, the uplifting of the PBL is slow because of the stable atmosphere.

At XH, the $CO_2$ mole fractions at L1 and L2 are closer to each other as compared to the two-layers measurements at BJ, because the difference in the vertical distance of two layers at XH is only 20 m. Nevertheless, we can still observe that the peak of the $CO_2$ at L2 occurs in the early morning later than that at L1 as the $CO_2$ at the ground level moved upward with the increase in convective PBL, with a large difference in winter and a small difference in summer.

To compare the vertical distribution of $CO_2$ at BJ and XH, we calculate the $CO_2$ gradient ($\delta CO_2 = \frac{CO_{2,L1} - CO_{2,L2}}{Alt_{L2} - Alt_{L1}}$) (Figure 12c), The diurnal variations of $\delta CO_2$ at BJ and XH have a similar pattern: close-zero during the day and positive at night. The maximum $\delta CO_2$ can reach to 0.6 ppm/m at XH in 2018 August and 0.2 ppm/m at BJ in 2018 November. The larger height difference at BJ (120 m) as compared to XH (20 m) may contribute to the smaller $\delta CO_2$.

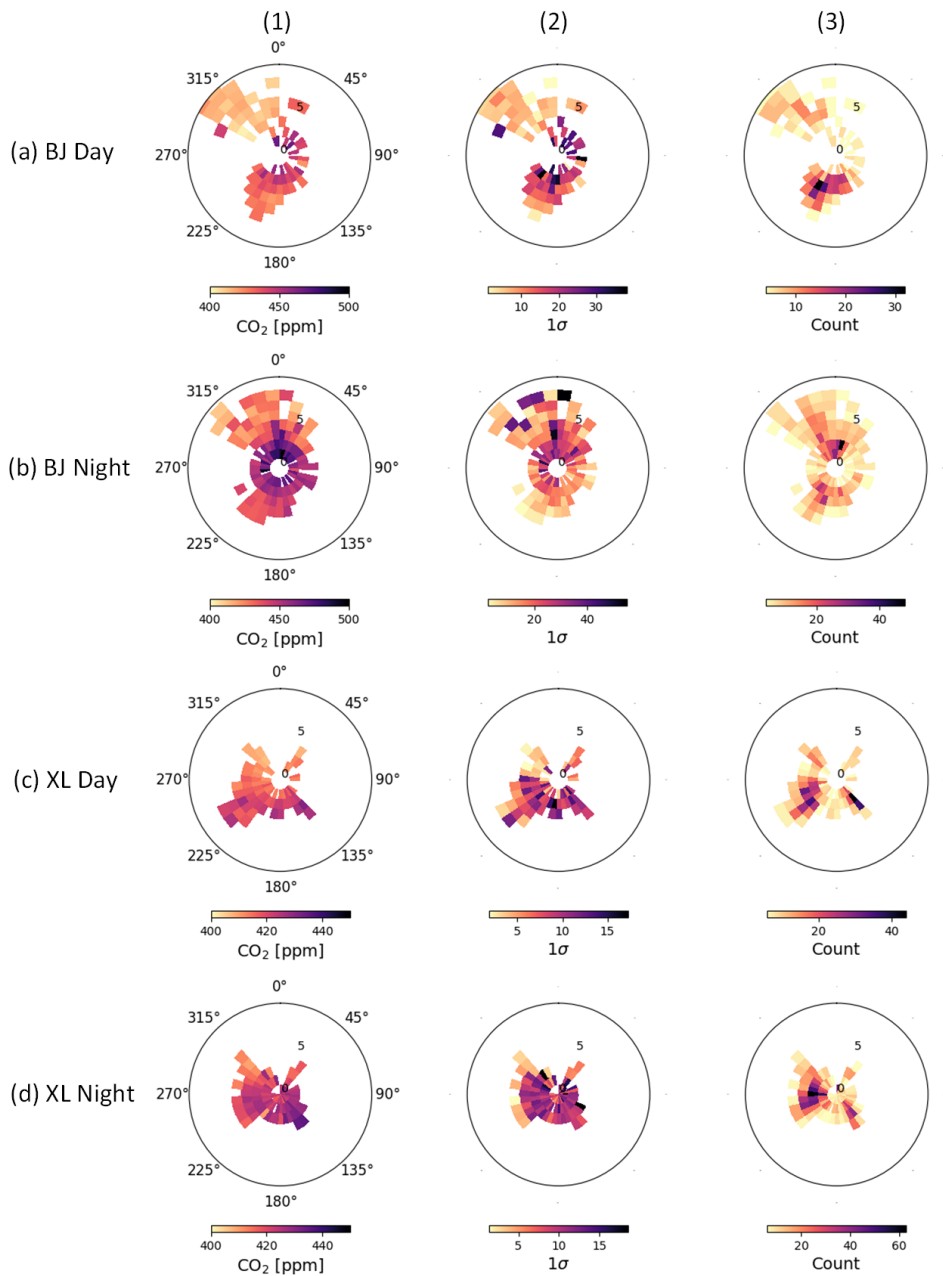

**Figure 11.** (1): binned $CO_2$ mole fraction as a function of wind speed $(m.s^{-1})$ and wind direction (°) at BJ L1 (a, b) and XL (c, d) based on daytime (14:00-16:00 LTC) and nighttime (22:00 -1:00 LTC) data between October 2018 and September 2019. (2): mean $1\sigma$ standard deviation of the $CO_2$ mole fractions in each bin. (3): the $CO_2$ measurement counts in each bin.

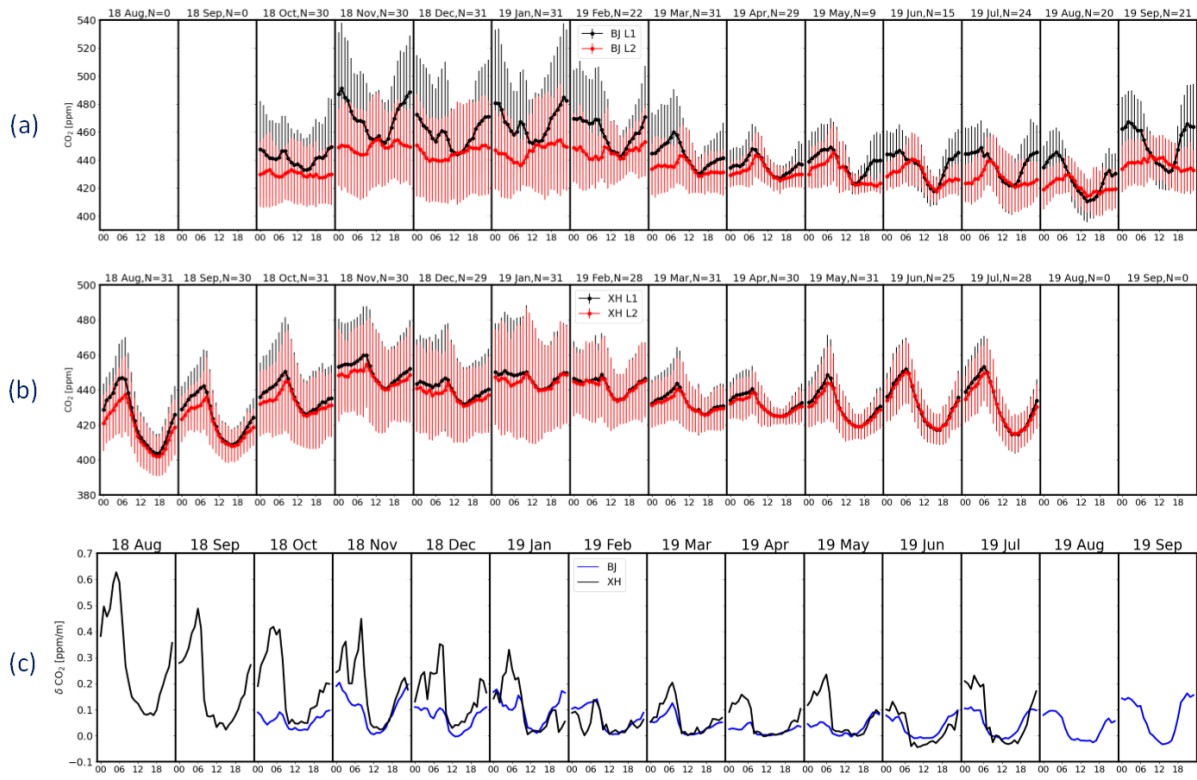

**Figure 12.** (a): the $CO_2$ measurements of BJ L1 and BJ L2 between October 2018 and September 2019. The error bars are the hourly standard deviations of $CO_2$. (b): the $CO_2$ measurements of XH L1 and XH L2 between August 2018 and July 2019. (c): the hourly $\delta CO_2$ [ppm/m] in each month at BJ and XH.

## 3.7 Weekday-weekend variation

Figure 13 shows the average hourly means of $CO_2$ on weekday, weekend and all days at BJ (L1), XH (L1), XL between October 2018 and September 2019, and BU (Boston) between April 2018 and April 2019. At BJ (L1), the nighttime $CO_2$ measurements on weekend from 20 pm to 6 am next morning are generally ~5 ppm larger than those on weekday. XH (L1) and XL $CO_2$ measurements on weekend are ~2 ppm than those on weekday throughout the whole day respectively. On the contrary, BU $CO_2$ measurements on weekday are ~8 ppm larger than those on weekend between 4 and 6 am. The $CO_2$ differences on weekday and weekend at BU turn smaller after sunrise. The mean $CO_2$ at BJ (L1), XH (L1), XL and BU is 447.6, 436.2, 420.3 and 429.8 ppm respectively on weekday, and 449.2, 437.6, 421.4 and 427.5 ppm respectively on weekend. The weekday-weekend variations at BJ and XH are similar to that at Nanjing China (Gao et al., 2018), where $CO_2$ mole fractions are higher on weekend, but different from Boston USA, London UK and Tamil Nadu India, where the $CO_2$ mole fractions are higher on weekday (Hernández-Paniagua et al., 2015; Kishore Kumar and Shiva Nagendra, 2015; Briber et al., 2013).

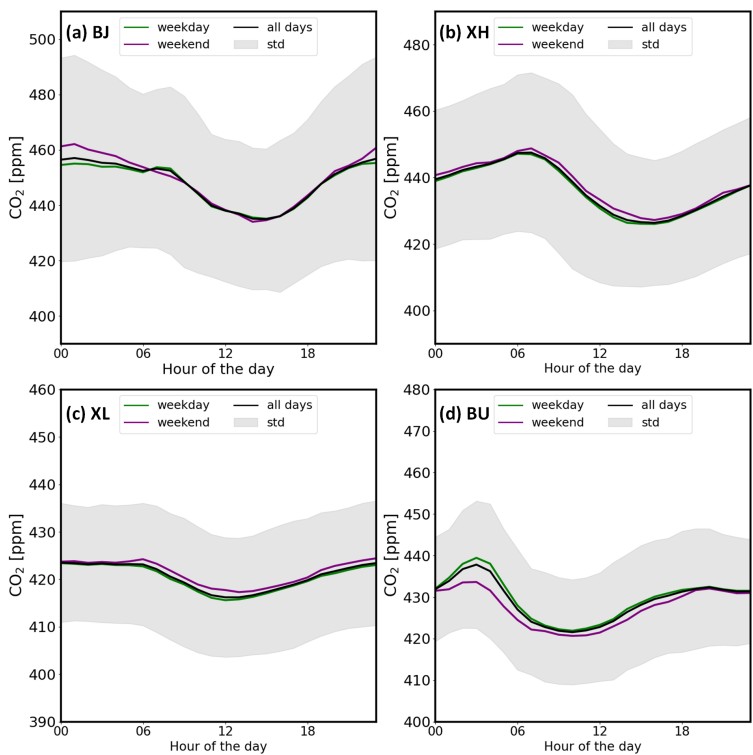

**Figure 13.** The average hourly means of $CO_2$ on weekday, weekend and all days at (a) BJ (L1), (b) XH (L1), (c) XL and (d) BU (Boston) between October 2018 and September 2019. The light gray shaded area represents one standard deviation from the mean for all days.

## 4    Conclusion

In this study, we show the $CO_2$ measurements from the in situ Picarro instruments at BJ, XH, and XL between June 2018 and March 2020. It is the first time to investigate $CO_2$ variations at these sites. BJ is inside the megacity, XH is in the suburban area, and XL is in the countryside on a mountain. The uncertainties of the $CO_2$ are 0.01, 0.06 and 0.02 ppm at BJ, XH and XL, respectively. The means and stds of $CO_2$ mole fractions are 448.4±12.8 ppm, 436.0±9.2 ppm and 420.6±8.2 ppm at BJ (L1), XH (L1) and XL, respectively. The CarbonTracker simulations at these three sites show fire, ocean and biogenic $CO_2$ are close

to each other throughout the whole year, and the variation of the fossil fuel $CO_2$ at XL is much less than that at BJ/XH. The $CO_2$ measurements at XL are used to represent the background and we find that there is a good relationship between the $CO_2$ enhancements at BJ and XH. BJ and XH are affected by $CO_2$ emissions and transports simultaneously. Comparison with other urban sites in US shows that the $CO_2$ concentration is the largest at BJ.

The variations of $CO_2$ at BJ, XH, and XL are discussed on diurnal and seasonal scales. It is found that the seasonal cycles

of $CO_2$ at these three sites are similar, with a high value in winter and a low value in summer, which is closely related to air temperature and LAI. However, the amplitudes of seasonal variations are different, with the values of 41.2 ppm, 36.1 ppm and

29.3 ppm at BJ, XH and XL, respectively. For the diurnal variation, the $CO_2$ is relatively low during the day and high at night. The diurnal variation of $CO_2$ at BJ, XH and XL is affected by the BLH, photosynthesis and human activities, and the impact of photosynthesis is more significant at XL.

The $CO_2$ measurements are compared against the local wind data at BJ and XL. At BJ, high $CO_2$ mole fractions are observed with low wind speeds ($< 2$ m.s$^{-1}$). At XL, the high $CO_2$ mole fractions during daytime are observed with the wind coming from the south, where the urban area is located.

    The two-levels measurements at BJ and XH show that the $CO_2$ mole fractions at lower and upper levels are close to each other during the day. The $CO_2$ mole fraction at the lower level is larger than that at the upper level at night with a vertical

gradient up to 0.6 ppm/m at XH and 0.2 ppb/m at BJ. The $CO_2$ mole fractions on weekend at BJ, XH and XL are found slightly higher than the ones on weekday.

*Author contributions.* MZ, TW, PW and GW desigend the experiment. YY performed the data curation. YY and MZ wrote the manuscript, and all authors read and provided comments on the paper.

*Competing interests.* The authors declare that they have no conflict of interest.

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
