# Peer review of "Spatial and temporal variations of CO2 mole fractions observed at Beijing, Xianghe and Xinglong in North China"

_Atmospheric Chemistry and Physics, 2021_

## Author Comment (AC2)

*Black: referee's comments red: authors' answers*
*First of all, we want to thank the referee 2 for the detailed analysis of our paper.*
*For the details, please look into the paper with keeping track of changes.*

Anonymous Referee #2

General comments: This is an important and valuable data to assess temporal and spatial variations of $CO_2$ in North China. However, this data must be further discussed in order to support the main reasons of those $CO_2$ variations. Comparisons with other megacities will be a good approach to improve the discussions.
Thanks for your suggestions.

In the revised paper, we add the $CO_2$ measurements at five urban sites in USA with a similar latitude of BJ. All these five sites belong to the $CO_2$ Urban Synthesis and Analysis (CO2-USA) Data Synthesis Network (Feng et al., 2016). The site locations, elevations, inlet heights, and references are listed in Table 1. As the $CO_2$ measurements at these five sites do not cover the period between October 2018 and September 2019, we use the latest 1-year available $CO_2$ measurements.

[Figure]

Figure 1. (a) Monthly means of $CO_2$ at BJ (L1), XH (L1), XL between October 2018 and September 2019, at BU, CRA, COM, IMC and SF during the latest 1 year and (b) the diurnal cycles of $CO_2$.

The monthly means and diurnal cycles of $CO_2$ at BJ (L1), XH (L1), XL, and 5 American urban sites are shown in Figure 1. It is found that the phases of the seasonal $CO_2$ cycles at BU, CRA, COM, IMC and SF are consistent with the observations at BJ (L1), XH (L1) and XL, with a high value in autumn-winter and a low value in summer. Among the five American sites, the highest $CO_2$ concentration is observed at IMC. The IMC site is inside a commercial zone and the $CO_2$ measurements over there are more strongly influenced by local emissions over there (Bares et al., 2019). The $CO_2$ concentration is also high at COM, because the Los Angeles megacity is one of the largest fossil fuel $CO_2$ emitters in the world (Matthäus et al., 2021). Figure 1 (a) shows that the $CO_2$ concentrations at COM and IMC are in the same level with the one at XH, but are less than the $CO_2$ concentration at BJ. The $CO_2$ concentrations at SF, BU and CRA are much lower as compared to BJ, because of lower anthropogenic emissions at these sites (McKain et al., 2015; Lauvaux et al., 2016; Shusterman et al., 2016).

Figure 1 (b) shows the diurnal variations of $CO_2$, with the amplitudes of 22.4, 19.4, 6.6, 16.3, 14.8, 41.5, 41.1 and 37.2 ppm at BJ (L1), XH (L1), XL, BU, CRA, COM, IMC and SF, respectively. The amplitudes of the diurnal variation at COM, IMC and SF are higher than that at BJ, although the yearly mean $CO_2$ levels at these sites are smaller than that at BJ. As the sampling heights at these sites and BJ are similar, the large amplitudes of the diurnal variation indicate that stronger variation in the local emissions and/or sinks exists at these three American sites as compared to BJ.

Table 1. Site characteristics of BJ, XH and XL in North China, BU, CRA, COM, IMC and SF in USA from the $CO_2$ Urban Synthesis and Analysis (CO2-USA) Data Synthesis Network.

| Site Code | Site Name | Lat (°N) | Lon (°E) | Elevation (m a.s.l.) | Inlet Height (m a.g.l.) | City | Reference |
|---|---|---|---|---|---|---|---|
| BJ | Beijing | 39.96 | 116.36 | 49 | 80/280 | Beijing | Cheng et al., 2018 |
| XH | Xianghe | 39.75 | 116.96 | 30 | 60/80 | Xianghe | Yang et al., 2020 |
| XL | Xinglong | 40.40 | 117.50 | 940 | 10 | Xinglong | Yang et al.,2019 |
| BU | Boston University | 42.35 | -71.10 | 4 | 29 | Boston | Sargent et al., 2018 McKain et al., 2015 |
| CRA | Crawfordsville | 39.99 | -86.74 | 264 | 76 | Indianapolis | Lauvaux et al., 2016 Richardson et al., 2017 |
| COM | Compton | 33.87 | -118.28 | 9 | 45 | Los Angeles | Verhulst et al., 2017 |
| IMC | Intermountain Medical Center | 40.67 | -111.89 | 1316 | 66 | Salt Lake City | Mitchell et al., 2018 Bares et al., 2019 |
| SF | SF Hospital Bldg 5 | 37.76 | -122.41 | 23.9 | 52 | San Francisco | Shusterman et al., 2016 |

Specific comments:

Section 2.2.2: Further description of calibration and data processing:

1) How CRDS stability was checked over time, before and after malfunctions?

Thanks for your suggestions. More information is added in the revised paper.

The intake system is connected to an 8-position valve, which is used to choose the air coming from the sample air, the target gas, or the calibration gas. The target and calibration gases are pressurized in 29.5 L treated aluminum alloy cylinders, which are scaled to the WMO X2007 standard by the China Meteorological Administration, Meteorological Observation Centre. The same calibration procedure is operated at these three sites: 1) 3-hours sample air; 2) 5-minutes calibration gas; 3) 3-hours sample air; 4) 5-minutes target gas. This process repeats every 6 hours and 10 minutes. Note that, the airs coming from two levels at XH and BJ are switched every 5 minutes during the 3-hours sample air period. As the remaining volume in the tubes needs time for flushing, the response of the analyzer turns to be stable about 1 minute after each switching. In order to reduce the uncertainty, we do not consider the first 3-minutes measurements after each switching.

The calibration gas is to calculate the calibration factor ($cf$),

$$cf = CO_{2,mcal}/CO_{2,cal} \tag{1}$$

where $CO_{2,mcal}$ is the $CO_2$ mole fraction measured by the Picarro analyzer from the calibration gas and $CO_{2,cal}$ is the standard $CO_2$ mole fraction of the calibration cylinder.

The target gas is used to check the precision and stability of the system. The T value are calculated as follows,

$$T = cf \times CO_{2,mtar} - CO_{2,tar} \tag{2}$$

where $CO_{2,tar}$ is the standard $CO_2$ mole fraction of the target gas cylinder, $CO_{2,mtar}$ is the $CO_2$ mole fraction measured by the Picarro analyzer from the target gas, $cf$ is calculated from the $CO_2$ mole fraction measured by the Picarro analyzer from the calibration gas.

To keep the CRDS stable over time, only the periods with T value within ±0.1 ppm are selected. The measurement uncertainties of the Picarro instrument at the three sites are calculated as the standard deviation (std) of T, which are 0.01, 0.06, and 0.02 ppm at BJ, XH, and XL respectively.

2) Describe the steps used during data processing; what kind of filters were used?

Besides the calibration procedure, we also do auto and manual flagging of the raw data. In each 1-hour $CO_2$ measurement window, auto-flags are assigned when deviations from $CO_2$ mean are found larger than 2-times hourly $CO_2$ std. Furthermore, manual flags are assigned by technicians at each site according to the logbook to exclude no-valid data resulted from the inlet filter, pump, and extreme weather issues. In addition, as the CRDS measurement system records $CO_2$ and $CH_4$ simultaneously, the variations of these two gases are checked together to manually flag $CO_2/CH_4$ outliers.

All these information has been added in the revised version.

P5 – lines 101-102: Data filtering were not used to reduce uncertainties but to exclude no-valid data. Review this sentence.

Thanks for the suggestion. The sentence is rewritten now as: "Furthermore, manual flags are assigned by technicians at each site according to the logbook to exclude no-valid data resulted from the inlet filter, pump, and extreme weather issues."

Results and discussion

Section 3.1 time series: Strategies/methods to selection of background mole fractions must be further presented and discussed in order to show low influence of anthropogenic sources.

Thanks for your suggestions.

In this study, we treat the $CO_2$ measurements at XL as the background of BJ and XH. In the revised paper, we use the CarbonTracker model, version CT-NRT.v2021-3 (Peters et al., 2005) to evaluate the influence of anthropogenic, biogenic, oceanic and fire sources at these three sites. The CarbonTracker is a data assimilation system developed by the National Oceanic and Atmospheric Administration (NOAA) to keep track of sources and sinks of atmospheric $CO_2$ around the world. Four tracers (biosphere, ocean, fire and fossil fuel) are treated separately to simulate atmospheric $CO_2$ mole fractions. Mustafa et al. (2020) evaluated the CarbonTracker model in Asia by comparing with satellite measurements, and they found that the CarbonTracker model captures the variation of $CO_2$ well. The model provides 3-hourly $CO_2$ data at 25 levels from surface to ~ 123 km, and the spatial resolution of the global CarbonTracker $CO_2$ simulation is 3°×2° (longitude x latitude). As BJ and XH are in the same model grid, we note the $CO_2$ simulations in the BJ/XH grid as BJ.

[Figure]

Figure 2. The time series of $CO_2$ simulations from fossil fuel ($CO_{2,ff}$), biosphere ($CO_{2,bio}$), fire ($CO_{2,fire}$) and ocean ($CO_{2,oce}$) modules at BJ/XH and XL.

Figure 2 shows the time series of $CO_2$ simulations from fossil fuel ($CO_{2,ff}$), biosphere ($CO_{2,bio}$), fire ($CO_{2,fire}$) and ocean ($CO_{2,oce}$) modules at BJ/XH and XL between October 2018 and September 2019. It is found that the fire and ocean $CO_2$ at BJ/XH and XL are close to each other throughout the whole year. The biogenic $CO_2$ at BJ/XH and XL have a similar level between October 2018 and June 2019, and become slightly different in summer 2019. However the difference in biogenic $CO_2$ is much less than that of the anthropogenic $CO_2$ differences. The variation of the fossil fuel $CO_2$ at XL is much less than that at BJ/XH. Therefore, by using the $CO_2$ measurements at XL as the background, we can significantly reduce the influence from fire, biosphere and ocean, and extract the signal of the anthropogenic $CO_2$ differences.

P7 - lines 138-144: Please add mean (std) concentrations related to higher and low $CO_2$ levels.
Done.
The mean $\Delta CO_2$ at BJ and XH are 26.2±20.6 ppm and 15.2±13.6 ppm, respectively.

P7 – lines 149-150: Contribution of main sources (fossil fuel and heating) must be further discussed. Other sources as biomass burning from wildfires are important? If possible, trace gases/species would be used to identify activity of specific sources.
Thanks for the suggestions. More discussions are added now.

The $\Delta CO_2$ has a maximum in winter and a minimum in summer at both BJ and XH. According to the CarbonTraker simulation, the high $CO_2$ concentrations at BJ and XH in winter are dominated

by the enhancement of fossil fuel. According to the Global Fire Assimilation System (GFAS) (https://www.ecmwf.int/en/forecasts/dataset/global-fire-assimilation-system) wildfire emissions, there is almost no biomass burning $CO_2$ emissions at BJ, XH and XL sites. The CarbonTraker model simulations confirm that fire $CO_2$ concentrations in this region are almost the same, and the simulated fire $CO_2$ at these sites are transported by the wildfire emissions at other places. What's more, the CarbonTraker model suggests that the fire $CO_2$ at these sites only take up a small proportion of the observed $CO_2$ (less than 5%).

P9 – lines 166-167: Discuss the reasons of higher amplitudes in BJ.
Thanks for the suggestions. More discussions are added now.

The amplitudes of the seasonal variation of $CO_2$ at BJ, XH and XL are 41.2 ppm, 36.1 ppm and 29.3 ppm, respectively. According to the CarbonTracker simulation, the $CO_2$ seasonal cycle in this region is mainly driven by the biogenic and anthropogenic $CO_2$. At XL, the anthropogenic $CO_2$ is almost constant through the whole year, while the biogenic $CO_2$ is low in summer and high in winter. For BJ/XH, apart from the similar biogenic $CO_2$ seasonal variation, the anthropogenic $CO_2$ is also high in winter and lower in summer. Therefore, combining the effect from the biosphere and human activities, the amplitude of $CO_2$ seasonal variation at BJ/XH is larger than that at XL. What's more, as the anthropogenic emission at BJ is much larger than that at XH, indicated by the EDGAR emission dataset, we thus observe the largest amplitude of the seasonal variation at BJ.

P9 – lines 216-217: References must be added to support the assumption.
Thanks for the suggestions. References are added now.

"The solar radiation is strongest at noon which leads to the largest photosynthesis removing $CO_2$ (Mohotti and Lawlor, 2002). The diurnal variation of $CO_2$ at daytime is then strongly affected by the plants in spring and summer, due to the large diurnal variation of the biogenic flux (high LAI) in these two seasons at XL (see Figure 3). However, in autumn and winter, the minimum of the $CO_2$ mole fraction occurs close to the maximum of the BLH, indicating that the diurnal variation is then dominated by the BLH (Newman et al., 2013), and the influence of the diurnal variation of the biogenic flux becomes less because of the low LAI in these two seasons."

[Figure]

Figure 3. The daily variations of biological $CO_2$ surface flux in spring, summer, autumn and winter at XL estimated from CarbonTracker.

Section 3.5: Reasons to $CO_2$ mole fractions variations in L1 and L2 altitudes must be discussed. Thanks for the suggestions, more discussions are added now.

Figure 10 in the ACPD shows the $CO_2$ hourly means observed at two levels at BJ and XH between October 2018 and September 2019. Note that, we select measurements when the hourly means are available at both levels.
At BJ, $CO_2$ mole fractions at L1 are generally higher than L2 as L1 is closer to near-ground human emissions. At BJ L1 (80 m a.g.l.), we can observe a peak in the early morning, which is corresponding to the transportation rush hour. The valley of $CO_2$ at BJ L1 occurs at 16:00-17:00 because of the maximum PBL resulting from the unstable atmosphere. After that, the atmosphere changes from unstable to stable during the night, leading to the $CO_2$ peak again. At BJ L2 (280 m a.g.l.), the diurnal variation of $CO_2$ generally follows that at L1. Note that the peak of the $CO_2$ at L2 occurs in the early morning later than that at L1 as the $CO_2$ at the ground level moved upward with the increase in convective PBL, with a large difference in winter and a small difference in summer. The $CO_2$ diurnal variations from two-layers Picarro measurements in 2018 and 2019 in our study are consistent with the seven open-path infrared gas analyzers (Model LI-7500A; at 8, 16, 47, 80, 140, 200 and 280 m a.g.l.) measurements between 2013 and 2016 at the same site (Cheng et al., 2018). In summer, the temperature is high due to a larger solar irradiance, the atmosphere becomes unstable quickly accelerating the uplifting of the PBL. In winter, the uplifting of the PBL is slow because of the stable atmosphere.
At XH, the $CO_2$ mole fractions at L1 and L2 are closer to each other as compared to the two-layers measurements at BJ, because the difference in the vertical distance of two layers at XH is only 20 m. Nevertheless, we can still observe that the peak of the $CO_2$ at L2 occurs in the early morning later than that at L1 as the $CO_2$ at the ground level moved upward with the increase in convective PBL, with a large difference in winter and a small difference in summer.
To compare the vertical distribution of $CO_2$ at BJ and XH, we calculate the $CO_2$ gradient ($\delta CO_2 =$

$(CO_{2,L1}-CO_{2,L2})/(Alt_{L2}-Alt_{L1}))$ (Figure 10c). The diurnal variations of $\delta CO_2$ at BJ and XH have a similar pattern: close-zero during the day and positive at night. The maximum $\delta CO_2$ can reach to 0.6 ppm/m at XH in 2018 August and 0.2 ppm/m at BJ in 2018 November. The larger height difference at BJ (120 m) as compared to XH (20 m) may contribute to the smaller $\delta CO_2$.

Section 3.6: This section must be further assessed using different approaches. One of these strategies would be investigate seasonal differences during weekday-weekend.
P13 – line 255: Assumption of lowest anthropogenic emissions on Tuesday must be proven.
Thanks for your suggestions.

From the Figure 11 in the ACPD, we see a low $CO_2$ concentration as compared to other days. Therefore, we wrote "we find that the $CO_2$ mole fractions have a minimum on Tuesday at BJ and XH. It is indicated that the anthropogenic emission is lowest on Tuesday." The most three important $CO_2$ anthropogenic emissions in BJ are energy conversion, transport and industry (Yu et al., 2014, Crippa et al., 2020). Among these three, only transport emission has a strong day-to-day variation. According to the TomTom report (https://www.tomtom.com/en_gb/traffic-index/beijing-traffic/), we can recognize a decrease in the transport during the weekend but not in Tuesday. Therefore, in the revised version, we removed the original Figure 11, but add the plots to compare the $CO_2$ during weekdays and weekends.

**🇨🇳 Beijing traffic**

**WEEKLY TRAFFIC CONGESTION BY TIME OF DAY**

What time was rush hour in Beijing?

| | Sun | Mon | Tue | Wed | Thu | Fri | Sat |
|---|---|---|---|---|---|---|---|
| 12:00 AM | 0% | 0% | 0% | 0% | 1% | 0% | 1% |
| | 0% | 0% | 0% | 0% | 0% | 0% | 0% |
| 02:00 AM | 0% | 0% | 0% | 0% | 0% | 0% | 0% |
| | 0% | 0% | 0% | 0% | 0% | 0% | 0% |
| 04:00 AM | 0% | 0% | 0% | 0% | 0% | 0% | 0% |
| | 0% | 0% | 0% | 0% | 0% | 0% | 0% |
| 06:00 AM | 1% | 12% | 8% | 8% | 8% | 7% | 0% |
| | 3% | 54% | 46% | 44% | 44% | 41% | 5% |
| 08:00 AM | 8% | 67% | 59% | 57% | 56% | 53% | 13% |
| | 13% | 58% | 52% | 50% | 49% | 48% | 20% |
| 10:00 AM | 16% | 38% | 37% | 37% | 36% | 36% | 26% |
| | 15% | 20% | 22% | 22% | 22% | 23% | 26% |
| 12:00 PM | 12% | 10% | 12% | 14% | 13% | 15% | 19% |
| | 11% | 10% | 12% | 13% | 13% | 15% | 14% |
| 02:00 PM | 18% | 13% | 16% | 17% | 17% | 19% | 19% |
| | 21% | 14% | 17% | 19% | 18% | 22% | 21% |
| 04:00 PM | 23% | 20% | 22% | 24% | 24% | 30% | 23% |
| | 29% | 43% | 46% | 47% | 47% | 53% | 30% |
| 06:00 PM | 29% | 52% | 54% | 55% | 55% | 61% | 31% |
| | 22% | 44% | 45% | 46% | 45% | 55% | 23% |
| 08:00 PM | 13% | 20% | 20% | 21% | 21% | 32% | 12% |
| | 8% | 10% | 11% | 12% | 12% | 22% | 7% |
| 10:00 PM | 4% | 6% | 7% | 8% | 8% | 14% | 4% |
| | 1% | 2% | 3% | 4% | 3% | 7% | 1% |

Figure 4. The traffic in Beijing within one week reported from TomTom.

[revised manuscript text omitted]

---

## Author Comment (AC3)

*Black: referee's comments red: authors' answers*
*First of all, we want to thank the referee 3 for the detailed analysis of our paper.*
*For the details, please look into the paper with keeping track of changes.*

Anonymous Referee #3

General comments: This is a carefully done study and the data is very valuable, but the preliminary data analysis and discussion have been done. What the main purpose of this study is? What's the main influencing mechanism of CO2? are there some differences with other big cites or megaregions?
Thanks for your suggestions.

The paper is aim to describe the spatial and temporal variations of $CO_2$ mole fractions in urban, suburban and rural areas of North China. And these measurements in and around cities are very useful for the climate and air pollution studies.
As is discussed in the paper, all the $CO_2$ variations at the three sites are influenced by the boundary layer height (BLH), photosynthesis and human activities. Generally, the increase of the BLH after sunrise and the photosynthetic uptake during the day make the $CO_2$ mole fraction decrease, but the BLH decreasing after sunset results into the accumulation of $CO_2$. However, at BJ, high $CO_2$ is more influenced by local human activities, and the $CO_2$ with the wind coming from the southwest is about ~21 ppm larger than those with the wind coming from the northwest during the day. At XL, besides the more significant impact of local photosynthesis, the wind from the cities in the south, such as Beijing and Tianjin, also makes $CO_2$ increase during the day.
In the revised paper, we add the $CO_2$ measurements at five urban sites in USA with a similar latitude of BJ. All these five sites belong to the $CO_2$ Urban Synthesis and Analysis (CO2-USA) Data Synthesis Network (Feng et al., 2016). The site locations, elevations, inlet heights, and references are listed in Table 1. As the $CO_2$ measurements at these five sites do not cover the period between October 2018 and September 2019, we use the latest 1-year available $CO_2$ measurements.

[Figure]

Figure 1. (a) Monthly means of $CO_2$ at BJ (L1), XH (L1), XL between October 2018 and September 2019, at BU, CRA, COM, IMC and SF during the latest 1 year and (b) the diurnal cycles of $CO_2$.

The monthly means and diurnal cycles of $CO_2$ at BJ (L1), XH (L1), XL, and 5 American urban sites

are shown in Figure 1. It is found that the phases of the seasonal $CO_2$ cycles at BU, CRA, COM, IMC and SF are consistent with the observations at BJ (L1), XH (L1) and XL, with a high value in autumn-winter and a low value in summer. Among the five American sites, the highest $CO_2$ concentration is observed at IMC. The IMC site is inside a commercial zone and the $CO_2$ measurements over there are more strongly influenced by local emissions over there (Bares et al., 2019). The $CO_2$ concentration is also high at COM, because the Los Angeles megacity is one of the largest fossil fuel $CO_2$ emitters in the world (Matthäus et al., 2021). Figure 1 (a) shows that the $CO_2$ concentrations at COM and IMC are in the same level with the one at XH, but are less than the $CO_2$ concentration at BJ. The $CO_2$ concentrations at SF, BU and CRA are much lower as compared to BJ, because of lower anthropogenic emissions at these sites (McKain et al., 2015; Lauvaux et al., 2016; Shusterman et al., 2016).

Figure 1 (b) shows the diurnal variations of $CO_2$, with the amplitudes of 22.4, 19.4, 6.6, 16.3, 14.8, 41.5, 41.1 and 37.2 ppm at BJ (L1), XH (L1), XL, BU, CRA, COM, IMC and SF, respectively. The amplitudes of the diurnal variation at COM, IMC and SF are higher than that at BJ, although the yearly mean $CO_2$ levels at these sites are smaller than that at BJ. As the sampling heights at these sites and BJ are similar, the large amplitudes of the diurnal variation indicate that stronger variation in the local emissions and/or sinks exists at these three American sites as compared to BJ.

Table 1. Site characteristics of BJ, XH and XL in North China, BU, CRA, COM, IMC and SF in USA from the $CO_2$ Urban Synthesis and Analysis (CO2-USA) Data Synthesis Network.

| Site Code | Site Name | Lat (°N) | Lon (°E) | Elevation (m a.s.l.) | Inlet Height (m a.g.l.) | City | Reference |
|---|---|---|---|---|---|---|---|
| BJ | Beijing | 39.96 | 116.36 | 49 | 80/280 | Beijing | Cheng et al., 2018 |
| XH | Xianghe | 39.75 | 116.96 | 30 | 60/80 | Xianghe | Yang et al., 2020 |
| XL | Xinglong | 40.40 | 117.50 | 940 | 10 | Xinglong | Yang et al.,2019 |
| BU | Boston University | 42.35 | -71.10 | 4 | 29 | Boston | Sargent et al., 2018 McKain et al., 2015 |
| CRA | Crawfordsville | 39.99 | -86.74 | 264 | 76 | Indianapolis | Lauvaux et al., 2016 Richardson et al., 2017 |
| COM | Compton | 33.87 | -118.28 | 9 | 45 | Los Angeles | Verhulst et al., 2017 |
| IMC | Intermountain Medical Center | 40.67 | -111.89 | 1316 | 66 | Salt Lake City | Mitchell et al., 2018 Bares et al., 2019 |
| SF | SF Hospital Bldg 5 | 37.76 | -122.41 | 23.9 | 52 | San Francisco | Shusterman et al., 2016 |

Specific comments:

1. Please explain the data processing method and the proportion of valid data at the three sites.

Thanks for your suggestions. More information about the data processing method is added in the revised paper.

(1) Calibration

The intake system is connected to an 8-position valve, which is used to choose the air coming from the sample air, the target gas, or the calibration gas. The target and calibration gases are pressurized

in 29.5 L treated aluminum alloy cylinders, which are scaled to the WMO X2007 standard by the China Meteorological Administration, Meteorological Observation Centre. The same calibration procedure is operated at these three sites: 1) 3-hours sample air; 2) 5-minutes calibration gas; 3) 3-hours sample air; 4) 5-minutes target gas. This process repeats every 6 hours and 10 minutes. Note that, the airs coming from two levels at XH and BJ are switched every 5 minutes during the 3-hours sample air period. As the remaining volume in the tubes needs time for flushing, the response of the analyzer turns to be stable about 1 minute after each switching. In order to reduce the uncertainty, we do not consider the first 3-minutes measurements after each switching.

The calibration gas is to calculate the calibration factor ($cf$),

$$cf = CO_{2,mcal}/CO_{2,cal} \qquad (1)$$

where $CO_{2,mcal}$ is the $CO_2$ mole fraction measured by the Picarro analyzer from the calibration gas and $CO_{2,cal}$ is the standard $CO_2$ mole fraction of the calibration cylinder.

The target gas is used to check the precision and stability of the system. The T value are calculated as follows,

$$T = cf \times CO_{2,mtar} - CO_{2,tar} \qquad (2)$$

where $CO_{2,tar}$ is the standard $CO_2$ mole fraction of the target gas cylinder, $CO_{2,mtar}$ is the $CO_2$ mole fraction measured by the Picarro analyzer from the target gas, $cf$ is calculated from the $CO_2$ mole fraction measured by the Picarro analyzer from the calibration gas.

To keep the CRDS stable over time, only the periods with T value within ±0.1 ppm are selected. The measurement uncertainties of the Picarro instrument at the three sites are calculated as the standard deviation (std) of T, which are 0.01, 0.06, and 0.02 ppm at BJ, XH, and XL respectively.

(2) Data Processing

Besides the calibration procedure, we also do auto and manual flagging of the raw data. In each 1-hour $CO_2$ measurement window, auto-flags are assigned when deviations from $CO_2$ mean are found larger than 2-times hourly $CO_2$ std. Furthermore, manual flags are assigned by technicians at each site according to the logbook to exclude no-valid data resulted from the inlet filter, pump, and extreme weather issues. In addition, as the CRDS measurement system records $CO_2$ and $CH_4$ simultaneously, the variations of these two gases are checked together to manually flag $CO_2/CH_4$ outliers.

The proportions of valid data are 98.5% and 99.1% at BJ L1 and L2, 99.3% and 99.1% at XH L1 and XH L2, 99.9% at XL.

All these information has been added in the revised version.

2. As $CO_2$ at XL is regarded as the background in this study, please explain whether there is a special data processing method for it, because the observational data at XL include not only the background information, but also local information about natural ecosystem and human activity, especially, the intake system of XL is on the roof.

Thanks for your suggestions.

In this study, we treat the $CO_2$ measurements at XL as the background of BJ and XH. In the revised paper, we use the CarbonTracker model, version CT-NRT.v2021-3 (Peters et al., 2005) to evaluate the influence of anthropogenic, biogenic, oceanic and fire sources at these three sites. The CarbonTracker is a data assimilation system developed by the National Oceanic and Atmospheric

Administration (NOAA) to keep track of sources and sinks of atmospheric $CO_2$ around the world. Four tracers (biosphere, ocean, fire and fossil fuel) are treated separately to simulate atmospheric $CO_2$ mole fractions. Mustafa et al. (2020) evaluated the CarbonTracker model in Asia by comparing with satellite measurements, and they found that the CarbonTracker model captures the variation of $CO_2$ well. The model provides 3-hourly $CO_2$ data at 25 levels from surface to ~ 123 km, and the spatial resolution of the global CarbonTracker $CO_2$ simulation is 3°×2° (longitude x latitude). As BJ and XH are in the same model grid, we note the $CO_2$ simulations in the BJ/XH grid as BJ.

[Figure]

Figure 2. The time series of $CO_2$ simulations from fossil fuel ($CO_{2,ff}$), biosphere ($CO_{2,bio}$), fire ($CO_{2,fire}$) and ocean ($CO_{2,oce}$) modules at BJ/XH and XL.

Figure 2 shows the time series of $CO_2$ simulations from fossil fuel ($CO_{2,ff}$), biosphere ($CO_{2,bio}$), fire ($CO_{2,fire}$) and ocean ($CO_{2,oce}$) modules at BJ/XH and XL between October 2018 and September 2019. It is found that the fire and ocean $CO_2$ at BJ/XH and XL are close to each other throughout the whole year. The biogenic $CO_2$ at BJ/XH and XL have a similar level between October 2018 and June 2019, and become slightly different in summer 2019. However the difference in biogenic $CO_2$ is much less than that of the anthropogenic $CO_2$ differences. The variation of the fossil fuel $CO_2$ at XL is much less than that at BJ/XH. Therefore, by using the $CO_2$ measurements at XL as the background, we can significantly reduce the influence from fire, biosphere and ocean, and extract the signal of the anthropogenic $CO_2$ differences.

3. It is very pity that there are no meteorological parameters at XH. For the situation (2.1) and the meteorological field (2.3), it seems the air masses from BJ can be captured much more at XH because "the percentage of wind frequency in the north region is 34%, 36%, 50% and 60%

respectively from spring to winter". And the air masses can be captured at XL only when the wind comes from SW.

Thanks for your the comment on this issue.

We are now devoting to fix the meteorology sensor at XH, which may provide more meteorological information in the future study.

References

[revised manuscript text omitted]